# Efficient termination of nuclear lncRNA transcription promotes mitochondrial genome maintenance

Dorine Jeanne Mariëtte du Mee[1], Maxim Ivanov[1], Joseph Paul Parker[1], Stephen Buratowski[2], Sebastian Marquardt[1]*

[1]Department of Plant and Environmental Sciences, Copenhagen Plant Science Centre, University of Copenhagen, Frederiksberg, Denmark; [2]Department of Biological Chemistry and Molecular Pharmacology, Harvard Medical School, Boston, United States

**Abstract** Most DNA in the genomes of higher organisms does not code for proteins. RNA Polymerase II (Pol II) transcribes non-coding DNA into long non-coding RNAs (lncRNAs), but biological roles of lncRNA are unclear. We find that mutations in the yeast lncRNA *CUT60* result in poor growth. Defective termination of *CUT60* transcription causes read-through transcription across the *ATP16* gene promoter. Read-through transcription localizes chromatin signatures associated with Pol II elongation to the *ATP16* promoter. The act of Pol II elongation across this promoter represses functional *ATP16* expression by a Transcriptional Interference (TI) mechanism. Atp16p function in the mitochondrial ATP-synthase complex promotes mitochondrial DNA stability. *ATP16* repression by TI through inefficient termination of *CUT60* therefore triggers mitochondrial genome loss. Our results expand the functional and mechanistic implications of non-coding DNA in eukaryotes by highlighting termination of nuclear lncRNA transcription as mechanism to stabilize an organellar genome.

DOI: https://doi.org/10.7554/eLife.31989.001

*For correspondence:
sebastian.marquardt@plen.ku.dk

**Competing interests:** The authors declare that no competing interests exist.

## Introduction

RNA Polymerase II (Pol II) transcribes protein-coding DNA sequences into mRNA. Pol II also transcribes many non-coding DNA (ncDNA) sequences into long non-coding RNA (lncRNA) molecules (*Ponting et al., 2009*). ncDNA sequences vastly outnumber protein coding DNA sequences in most eukaryotic genomes. Genome-wide Pol II activity at ncDNA is referred to as 'pervasive transcription' of eukaryotic genomes (*Jensen et al., 2013*). The purpose of pervasive transcription is currently unclear. The 'transcriptional noise' hypothesis postulates that widespread Pol II transcription of ncDNA is a by-product of promiscuous Pol II activity and may often not be functionally relevant (*Struhl, 2007*). Indeed, the expression levels of lncRNA are generally lower than those of genes (*Schlackow et al., 2017*). Reasons for low expression levels of lncRNA include a combination of tissue-specific expression and targeted lncRNA degradation (*Deveson et al., 2017*; *Jensen et al., 2013*).

Measurements of nascent Pol II transcription reveal that lncRNA transcription can be strong, for example divergent lncRNA transcription from gene promoters (*Churchman and Weissman, 2011*). The strength of divergent lncRNA transcription is actively regulated at the level of initiation through chromatin-based mechanisms (*Marquardt et al., 2014*). Profiling RNA expression in RNA decay pathway mutants supports the view that nascent lncRNA transcription is often stronger than expected based on steady-state RNA measurements in wild type (*Jensen et al., 2013*). The regulation of lncRNA levels through targeted RNA decay and at the level of initiation argue that lncRNA

**eLife digest** Genes are made up of DNA and contain the information to make proteins, which carry out a variety of roles in the cell and the body. First, the information found on DNA needs to be transcribed into RNA molecules, which then act as a template to build the actual proteins. However, the vast majority of DNA does not encode proteins. Nevertheless, these non-coding regions of DNA (often given the popular but misleading name 'junk-DNA') are still transcribed into non-coding RNA. The purpose of this type of RNA is largely unclear, although some are known to activate certain genes. The transcription of non-coding RNA is also sensitive to environmental changes, suggesting it may play other important roles in the cell.

Not all DNA – including non-coding DNA– is copied into RNA in one go. Usually, every DNA sequence is transcribed separately as one unit. These units have clearly marked start and end points. If these marker points are overridden the transcription process can overlap onto the next sequence. Thus, in the case of coding DNA, proteins may not form properly. However, until now it was unclear if missed marker points in non-coding RNA may also have consequences.

To investigate this further, du Mee et al. mutated several non-coding parts of the DNA in yeast. The experiments showed that a non-coding RNA sequence called CUT60, appeared to be important to help yeast cells grow. When CUT60 was modified so that it lacked the end marker, its RNA transcript fused with the neighbouring gene called ATP16. As a result, the protein of the ATP16 gene could no longer be produced properly. Normally, ATP16 plays important roles in a cell structure called the mitochondrion, also known as the energy powerhouse of the cell. The mitochondrion has its own DNA, and without CUT60 and ATP16, the yeast cells lost their mitochondrial DNA and could not grow as quickly.

This shows that non-coding DNA sequences can have a purpose and can affect other parts of the cell. Moreover, start and end markers of transcription are also important in non-coding DNA sequences. The same mechanism could be at play in other genes or even other organisms. As well as revealing a new role for non-coding DNA, the findings could also help to develop a new method to cleanse yeast cells of disease-causing mutations in their mitochondrial DNA.

DOI: https://doi.org/10.7554/eLife.31989.002

levels in cells are carefully balanced. However, some intensely characterized human lncRNA such as *H19*, *Xist*, *NORAD* and *Malat1* represent relatively stable lncRNA molecules (*Schlackow et al., 2017*). As a class, lncRNA molecules are poorly conserved on sequence level, which challenges the idea of an evolutionarily conserved function of lncRNA (*Graur et al., 2015*). However, additional considerations such as secondary RNA structure, splicing or genomic location may be important factors when considering lncRNA conservation (*Seemann et al., 2017*; *Ulitsky, 2016*). As some lncRNA molecules can serve biological roles, the distinction between functional lncRNA transcription and transcriptional noise requires experimental testing. Progress on this question is aided by a community consensus on the experiments needed to assign functions to ncDNA (*Bassett et al., 2014*; *Goff and Rinn, 2015*).

In budding yeast, profiling transcripts in RNA decay pathway mutants reveals many classes of 'cryptic' lncRNA molecules that accumulate specifically in these mutants. Cryptic Unstable Transcripts (CUTs) accumulate in mutants of the nuclear exosome 3'-to-5' RNA decay pathway (*Davis and Ares, 2006*; *Wyers et al., 2005*; *Xu et al., 2009*). Mutations in the 5'-to-3' cytoplasmic decapping-based RNA degradation pathway define Xrn1-sensitive Unstable Transcripts (XUTs) (*van Dijk et al., 2011*). Stable Uncharacterized Transcripts (SUTs) were initially defined based on their environment-specific expression pattern (*Xu et al., 2009*). SUTs are targeted for degradation by both the nuclear exosome and the cytoplasmic Xrn1 pathway (*Marquardt et al., 2011*). Even though lncRNAs classified into the same subgroup based on RNA decay pathway sensitivity reveal some common characteristics, it is currently unclear to what extent this classification indicates similar functions.

Processes linked to the act of Pol II transcription play an important role in targeting RNA decay pathways. Different stages in the 'transcription cycle' model of Pol II transcription are defined by the localization of molecular hallmarks that instruct stage-specific Pol II activities (*Buratowski, 2009*).

The essential budding yeast Nrd1-Nab3-Sen1 (NNS) pathway mediates termination of Pol II transcription of short non-coding transcripts including CUTs and SUTs (*Porrua and Libri, 2015*; *Schulz et al., 2013*). A combination of three molecular hallmarks associated with early stages of Pol II transcription recruit NNS to nascent lncRNA transcripts: Histone 3 lysine 4 trimethylation (H3K4me3), RNA consensus motifs in nascent lncRNA, and phosphorylation of the serine five residue in the Pol II largest subunit C-terminal repeat domain (Pol II CTD) (*Carroll et al., 2007*; *Terzi et al., 2011*; *Vasiljeva et al., 2008*). Interestingly, phosphorylation of the Pol II CTD at tyrosine 1 has the opposite effect and decreases NNS recruitment (*Yurko et al., 2017*). NNS-mediated transcriptional termination of CUTs is directly linked to the recruitment of the nuclear exosome RNA degradation machinery (*Vasiljeva and Buratowski, 2006*). While NNS components are specific to budding yeast, equivalent pathways ensure that early transcription termination of lncRNA is linked to co-transcriptional degradation by the nuclear exosome in mammals (*Ntini et al., 2013*; *Ogami et al., 2017*; *Preker et al., 2008*). lncRNA abundance in cells is therefore determined by molecular checkpoints operating co-transcriptionally that are tightly linked to the termination of Pol II transcription.

The termination of Pol II transcription for polyadenylated RNAs is currently best described by the 'Torpedo Model' (*Kim et al., 2004*; *Richard and Manley, 2009*; *West et al., 2004*). This mechanism relies on the cleavage of the nascent transcript. The continuing polymerase, which is transcribing beyond the mature transcript end site, is chased by 5'-to-3' exonucleases that dislodge Pol II from the DNA template. The system relies on the identification of termination specifying RNA sequences by polyadenylation/termination factors, transcription stage specification, and reducing the velocity of the polymerase after transcript cleavage. Therefore, gene expression relies on efficient termination of Pol II transcription, for example through tight linkage of termination and mRNA polyadenylation (*Proudfoot, 2016*).

Failures in the Pol II termination process can impact on genomic stability (*Aguilera and García-Muse, 2012*). Inefficient termination can blur the boundaries of neighboring transcription units with negative effects on gene expression (*Ard et al., 2017*). Examples of this phenomenon have been initially described through studies of two α-globin gene copies oriented in tandem and termed 'Transcriptional Interference' (TI) (*Proudfoot, 1986*). TI of gene promoters by read-through transcription of upstream protein coding genes has since been detected in additional organisms (*Greger et al., 2000*; *Hedtke and Grimm, 2009*). TI repression of gene promoters can also be triggered by overlapping lncRNA transcription (*Ard et al., 2014*; *Kim et al., 2016*; *Mellor et al., 2016*; *Touat-Todeschini et al., 2017*), or by termination failures of transcripts embedded within lncRNA such as pri-miRNA (*Dhir et al., 2015*). Overall, these studies support an important role of Pol II termination to maintain genomic integrity and gene expression.

Upstream lncRNA transcription across the downstream gene promoter represents a mechanism to adjust gene expression to the environmental conditions in budding yeast (*Martens et al., 2004*). Even though Pol II is actively transcribing across the promoter of the downstream *SER3* gene required for serine biosynthesis, no initiation from the *SER3* promoter occurs through co-transcriptional changes in the chromatin structure associated with Pol II elongation (*Hainer et al., 2011*; *Martens et al., 2004*). Similarly, gene expression can be attenuated by depositing methylation marks over promoters by cryptic transcription (*Kim et al., 2012*; *Pinskaya et al., 2009*). Regulation of gene expression by TI is intriguing, because regulation by it could reconcile equivalent functions of lncRNA that are expressed at low levels and are poorly conserved at sequence level.

Here, we find that transcriptional termination of the lncRNA *CUT60* in budding yeast insulates the downstream gene from interfering transcription resulting from promoter bi-directionality. Efficient termination of *CUT60* promotes expression of the downstream *ATP16* by preventing TI. Reduced *ATP16* expression causes poor growth and triggers the loss of mitochondrial DNA (mtDNA). Our findings highlight the efficiency of nuclear lncRNA termination as a mechanism to promote functional gene expression and mtDNA stability.

## Results

### *CUT60* is required for mitochondrial DNA maintenance

The lncRNA *CUT60* is divergently transcribed from the promoter of the mediator complex subunit *MED2* (*Poss et al., 2013*). In addition, *CUT60* is located upstream in tandem of the *ATP16* gene

encoding the δ-subunit of the mitochondrial ATP-synthase that couples proton translocation to ATP synthesis (*Duvezin-Caubet et al., 2003*; *Giraud and Velours, 1994*) (*Figure 1A*). Native Elongating Transcript sequencing (NET-seq) data (*Marquardt et al., 2014*) indicates higher nascent Pol II transcription of the two flanking genes than of *CUT60* (*Figure 1—figure supplement 1A*). To test if this low level of *CUT60* transcription may be functionally significant, we replaced *CUT60* with the *URA3* coding sequence (cut60Δ::URA3) (*Figure 1—figure supplement 1B–C*). We detect growth on selective media arguing for functional transcription (*Figure 1—figure supplement 1D*). We note that multiple independently generated isolates of cut60Δ::URA3 mutants had a strongly reduced growth rate compared to wild type (*Figure 1B*, *Figure 1—figure supplement 1E*). Common growth defects in multiple isolates argue against a second-site mutation. Reduced growth may be attributed to reduced mitochondria function. To test mitochondria function, we assayed yeast growth on non-fermentable carbon sources such as ethanol and glycerol. Strikingly, cut60Δ::URA3 fails to grow on non-fermentable carbon sources, indicating mitochondrial defects (*Figure 1C*, *Figure 1—figure supplement 1F*). To test lesions in the mitochondrial genome, we performed test crosses using cut60Δ::URA3 to wild type and the mip1Δ mutant that has lost the mitochondrial genome (i.e. genotype $\rho^0$) through DNA replication defects (*Foury, 1989*). As mitochondria are inherited through the cytoplasm, the resulting diploids can grow on non-fermentable carbon sources if one of the parents contains functional mitochondria (*Merz and Westermann, 2009*). We find that diploids resulting from crosses of cut60Δ::URA3 with mip1Δ fail to grow on non-fermentable carbon sources, while they show growth on the fermentable carbon source glucose (*Figure 1D*). These data strongly support that cut60Δ::URA3 mutants have mitochondrial defects. To test if the cut60Δ::URA3 mutation may result in a reduction ($\rho^-$) or loss ($\rho^0$) of mitochondrial DNA, we measured mitochondrial DNA using

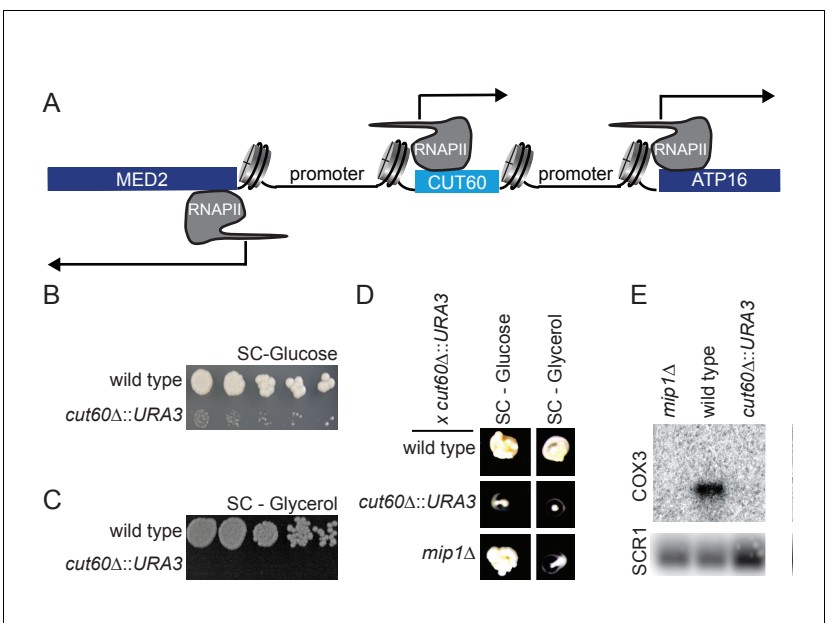

**Figure 1.** The divergent non-coding transcript *CUT60* promotes growth and mitochondrial genome maintenance. (**A**) Schematic representation of the bidirectional promoter of *MED2*. *CUT60* is a divergent lncRNA transcript originating from the *MED2* promoter, and it is located upstream in tandem of *ATP16*. (**B**) Serial dilution growth assay of wild type and cut60Δ::URA3 strains on SC-Glucose medium (n = 3). (**C**) Serial dilution growth assay of wild type and cut60Δ::URA3 strains on SC-Glycerol medium (n = 3). (**D**) Growth of wild type, cut60Δ::URA3 and mip1Δ on SC-Glucose and SC-Glycerol after mating with cut60Δ::URA3 strain (n = 3). (**E**) Northern blot analysis of *COX3* and *SCR1* (as loading control) transcripts in mip1Δ, wild type and cut60Δ::URA3 (n = 3).

DOI: https://doi.org/10.7554/eLife.31989.003

The following figure supplement is available for figure 1:

**Figure supplement 1.** NET-seq data showing low transcription from *CUT60* locus compared to *MED2* and *ATP16* coding regions.

DOI: https://doi.org/10.7554/eLife.31989.004

quantitative PCR with probes specific to the mitochondrial genome (*Figure 1—figure supplement 1G*). We find that the mitochondrial DNA level in *cut60Δ::URA3* and *mipΔ1* controls are reduced compared to wild type yeast. We detect no qPCR signal corresponding to mtDNA, suggesting a $\rho^0$ phenotype in *cut60Δ::URA3*. If *cut60Δ::URA3* mutants were $p^0$, we would expect that expression of transcripts encoded on the mitochondrial genome cannot be detected. To test this hypothesis, we assayed expression of the highly abundant *COX3* transcript encoded by the mitochondrial genome (*Figure 1E*). While we can detect strong *COX3* expression in wild type, we fail to detect any *COX3* expression in the $p^0$ control mutant *mip1Δ* and *cut60Δ::URA3*. Remarkably, our collective results suggest the loss of mtDNA when the *CUT60* lncRNA sequence of the nuclear genome is disrupted.

## *CUT60* functions in cis to maintain functional *ATP16* expression

To understand how *CUT60* stabilizes the mitochondrial genome, we hypothesized that *CUT60* may affect the expression of genes involved in mtDNA maintenance. Mitochondrial genome loss can be triggered in cells lacking Atp16p (*Duvezin-Caubet et al., 2006*). ATP-synthase proton channel formation requires multiple genes located on the mitochondrial genome; therefore the loss of mtDNA represents a strategy to prevent uncoupled proton translocation from ATP synthesis. We hypothesized that the observed mitochondrial genome loss could be triggered if mutating *CUT60* affected expression of neighboring *ATP16* (*Duvezin-Caubet et al., 2006*). To test *cis*-acting functions of *CUT60* we analyzed expression of the flanking genes. Expression of the downstream *ATP16* is strongly reduced in *cut60Δ::URA3* mutants, while we detect no expression changes of the *MED2* gene (*Figure 2A*, *Figure 2—figure supplement 1A*). To enhance detection of low-level or unstable transcripts, we included the 5'-to-3' RNA decay pathway mutant *xrn1Δ*, and the 3'-to-5' nuclear exosome RNA decay pathway mutant *rrp6Δ* in our analysis (*Houseley and Tollervey, 2009*). We noticed a different size profile of *ATP16* RNA in *cut60Δ::URA3* mutant strains with RNA degradation defects through *rrp6Δ* or *xrn1Δ* (*Figure 2A*). While we detect no *ATP16* mRNA expression of the expected size in the mutants, we observe an extended *ATP16* transcript in *rrp6Δ*, and at higher levels in *xrn1Δ*. The size of the extended *ATP16* transcript is approximately (1.9 kb), which would be in agreement with a hypothesized *URA3-ATP16* fusion transcript. To test if the extended *ATP16* transcript represents a 5'-extension, we used northern blotting probes against *URA3* and the ATP16 promoter region downstream of the annotated *CUT60* 3'-end (*Figure 2—figure supplement 1B*). Both probes detect a transcript coinciding in size with the extended *ATP16* transcript. This data suggest that the extended *ATP16* transcript detected in the *cut60Δ::URA3* mutant represents 5'-extension that originates from the *CUT60* promoter and fails to efficiently terminate, resulting in read-through transcription into *ATP16*. A similar repression of *ATP16* is observed in mutants replacing *CUT60* with *SUT129* (*cut60Δ::SUT129*), indicating that this effect is not dependent on the *URA3* sequence (*Figure 2A*). *CUT60* can be expressed from the *SUT129* regulatory region (*sut129Δ::CUT60*) with CUT-specific expression characteristics (*Marquardt et al., 2014*). However, expressing *CUT60 in trans* from the *SUT129* regulatory regions (*sut129Δ::CUT60*) does not restore *ATP16* expression (*Figure 2A*). These data indicate a *cis*-acting function of the *CUT60* locus to promote *ATP16* expression, either as DNA element or non-coding transcript.

ATP16 expression could rely on DNA sequence elements located in *CUT60* ('DNA element model'), or from *CUT60* lncRNA function by a *cis*-acting mechanism ('*cis*-acting lncRNA model'). To distinguish these two models, we cloned four versions of *ATP16* expression constructs with variable 5' extensions in yeast expression plasmids and transformed them into *cut60Δ::URA3* (*Figure 2B*). We measured *ATP16* expression derived from plasmids by northern blotting and compared expression to wild type containing the empty vector control (*Figure 2C*). We quantified *ATP16* expression relative to the *SCR1* loading control to test if the constructs restore *ATP16* expression to wild-type level (*Figure 2D*). Our results indicate that all constructs restore *ATP16* expression to the level detected in wild type. A comparison of *ATP16* expression between construct V1 (lacking *CUT60*) and construct V2 (including *CUT60*) most directly tests the effect of potential DNA *cis*-elements located within the *CUT60* sequence. Our statistical tests show no significantly different *ATP16* expression between these constructs. These data strongly argue against DNA sequence elements providing promoter function located in *CUT60* as explanation for reduced *ATP16* expression and growth in cells lacking *CUT60* sequences. We note that in the only construct lacking *CUT60* sequences a second, longer transcript accumulates (see discussion). Even though *ATP16* expression can be restored using constructs V1-4, a growth defect compared to wild type remains. Interestingly,

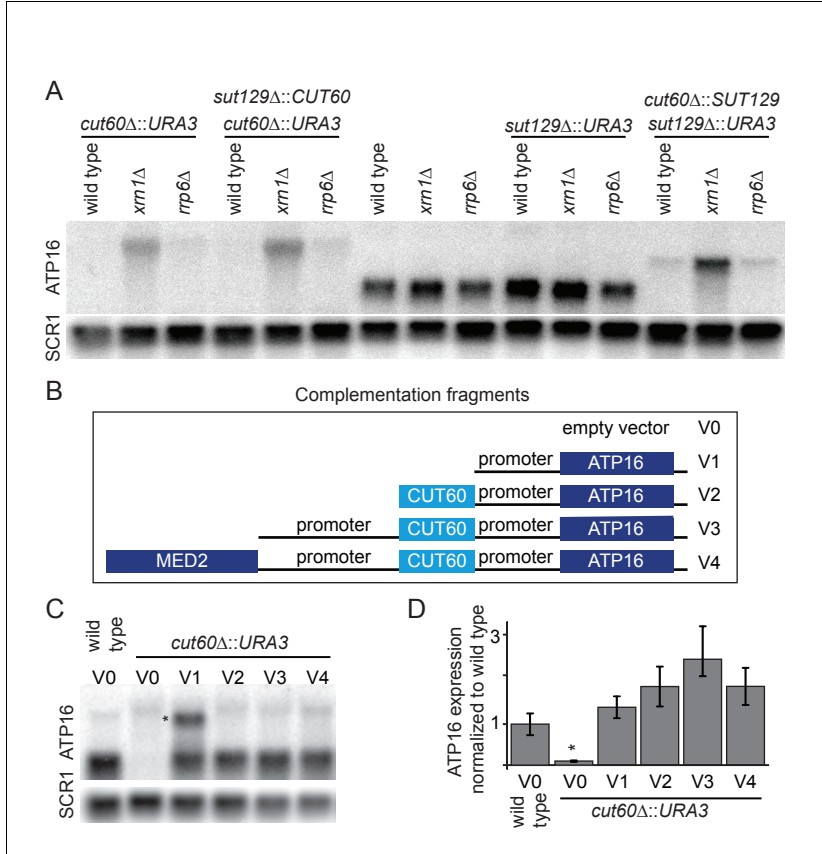

**Figure 2.** *CUT60* promotes *ATP16* expression as cis-acting transcript. (**A**) Northern blot analysis of *ATP16* and *SCR1* (as loading control) transcripts in *cut60Δ::URA3*, *sut129Δ::CUT60 + cut60Δ::URA3*, wild type, *sut129Δ::URA3* and *cut60Δ::SUT129 + sut129Δ::URA3* strains. For all strains the transcripts were analyzed in three different backgrounds: Wild type or respective decay pathway mutant background indicated above blot (n = 1). (**B**) Schematic representation of the five different fragments used for complementation in *Figure 2C*. V0 is the empty vector control, V1 contains the *ATP16* promoter sequence excluding *CUT60*, V2 expands V1 with *CUT60*, V3 expands V2 with the *MED2* promoter, V4 expands V3 with the sequence of *MED2*. (**C**) Northern blot analysis of *ATP16* and *SCR1* (as loading control) transcript in wild type +empty vector (V0, as shown in *Figure 2B*), *cut60Δ:: URA3* with complementation vectors (V0-4) (n = 3). * indicates extended transcript in *cut60Δ::URA3* + V1 strain. (**D**) Quantification of *ATP16* expression of *Figure 2C*. Error bars are s.e.m. of three biological replicates. Statistical significance assessed by t-test, wild type compared to *cut60Δ::URA3* is significant (p<0.02). All complementation vectors show no statistical significant increase of *ATP16* expression compared to wild type *ATP16* expression.
DOI: https://doi.org/10.7554/eLife.31989.005

The following figure supplement is available for figure 2:

**Figure supplement 1.** *CUT60* influences downstream *ATP16* locus in cis.
DOI: https://doi.org/10.7554/eLife.31989.006

---

growth in strains carrying the constructs is slightly improved compared to *cut60Δ::URA3* mutants (*Figure 2—figure supplement 1C*). The growth defects are consistent with mitochondrial genome loss: once mtDNA is lost, reconstitution of *ATP16* expression is insufficient to restore wild type growth. In conclusion, our findings suggest a *cis*-acting function for the *CUT60* lncRNA to promote functional *ATP16* expression, which is necessary to maintain the mitochondrial genome.

## *ATP16* expression requires efficient upstream transcriptional termination

lncRNA species are commonly defined based on expression profiling in characteristic mutant backgrounds, but it remains unclear if these lncRNA definitions carry functional implications. The *cis*-acting function of *CUT60*, in combination with strengths of our experimental system, allows us to

address this question. To test if the function of *CUT60* can be replaced by other lncRNA in the same genomic location as *CUT60* we generated replacement mutants by site-specific homologous recombination and assayed *ATP16* expression. To test if a lncRNA of the SUT class can restore *ATP16* expression, we replaced *CUT60* with *SUT129* (*cut60Δ::SUT129*) (*Xu et al., 2009*). We removed endogenous *SUT129* (*sut129Δ::URA3*) to avoid potentially confounding effects in the same background (*Figure 3A*). In *cut60Δ::SUT129*, an extended *ATP16* transcript with reduced expression is detected (*Figure 3A*). We conclude that *SUT129* cannot substitute *CUT60* function.

Transcriptional termination of SUTs is often inefficient, resulting in 3'-extended SUTs (eSUTs) (*Marquardt et al., 2011*). This suggests that a SUT lncRNA might promote *ATP16* expression if it was terminated like a CUT. To test this hypothesis, we generated *CUT60-SUT129* hybrids by fusing these sequences in the transcript middles (*Figure 3—figure supplement 1A–B*). Expressing the *5'-CUT60-3'-SUT129* hybrid instead of *CUT60* (*cut60Δ::5'C-3'S*) resulted in low *ATP16* expression as part of an extended transcript as in the *cut60Δ::URA3* mutant (*Figure 3A*). We note the size of *ATP16*-containing fusion transcript in *cut60Δ::5'C-3'S* is slightly reduced compared with *cut60Δ::SUT129*, consistent with read-through transcription of *SUT129* 3'-end into *ATP16*. Strikingly, *ATP16* expression is restored when the *5'-SUT129-3'-CUT60* hybrid replaces *CUT60* (*Figure 3A*, *Figure 3—figure supplement 1C*). Moreover, a short transcript with SUT-specific accumulation profile in RNA decay mutants can be detected with a probe against the *SUT129* 5'-region (*Figure 3—figure supplement 1D*). The detection of a *SUT*-like transcript resulting from the *5'-SUT129-3'-CUT60* hybrid maps efficient transcriptional termination to the 3'-half of *CUT60*. These data suggest that efficient transcriptional termination of *CUT60* promotes *ATP16* expression.

The essential Nrd1-Nab3-Sen1 (NNS) Pol II transcription termination pathway terminates CUT transcription (*Mischo and Proudfoot, 2013*). A role for NNS-mediated transcriptional termination of *CUT60* is supported by hypomorphic *nrd1* mutations that reveal 3'-extended *CUT60* RNA species (*Marquardt et al., 2011*). Nascent transcription data (Pol II PAR-CLIP) following conditional Nrd1 depletion support a direct role of Nrd1 in *CUT60* termination as depletion results in more Pol II-associated RNA downstream of *CUT60* (*Schaughency et al., 2014*) (*Figure 3—figure supplement 2A*). To test if transcription termination defects in *CUT60* resulting from mutations in trans-acting factors share phenotypes observed in *cut60Δ* mutations, we assayed growth of two hypomorphic *nrd1* mutant alleles in different genetic backgrounds, *nrd1-1* and *nrd1Δ151–214* (*Steinmetz and Brow, 1998*; *Vasiljeva et al., 2008*) (*Figure 3B*, *Figure 3—figure supplement 1E*). Strikingly, *nrd1-1*, *nrd1Δ151–214* and the *mip1Δ* $\rho^0$ control fail to grow on non-fermentable carbon sources, while we can recapitulate the reduced growth phenotype of both mutant alleles on fermentable carbon sources. These data show that *nrd1-1* and *nrd1Δ151–214* mimic growth defects of *cut60* mutations. The shared growth defects suggest that *ATP16* expression may be reduced when Nrd1 function and thus NNS termination is impaired. We assayed *ATP16* expression by northern blotting to test the effects of *nrd1-1* and *nrd1Δ151–214* (*Figure 3C*). Both *nrd1* mutant alleles reduce *ATP16* expression by about 40% compared to their respective isogenic controls (*Figure 3—figure supplement 1F*). While this expression is higher compared to *cut60Δ* mutations (*Figure 2*), we note that the hypomorphic *nrd1* mutant alleles provide residual Nrd1 termination function. Moreover, it has been estimated that reducing *ATP16* by around 50% is sufficient to trigger mtDNA loss (*Duvezin-Caubet et al., 2003*). In conclusion, these data suggest that *CUT60* termination by the NNS pathway promotes *ATP16* expression and mtDNA maintenance.

NNS-mediated termination is tightly connected to the classification of budding yeast CUTs as NNS-termination is linked to the characteristic RNA degradation through the nuclear exosome pathway (*Arigo et al., 2006*; *Fox and Mosley, 2016*; *Porrua and Libri, 2015*; *Vasiljeva and Buratowski, 2006*). To test if other CUTs can functionally substitute for *CUT60*, we screened CUTs based on their genomic location, and selected CUTs located upstream in tandem of a protein coding gene along with distance filtering (>0.1 kb;<1.5 kb) (*Xu et al., 2009*). Furthermore, we excluded CUTs with transcripts annotated on either strand between the CUT and the downstream gene, resulting in a list of 68 CUTs (*Figure 3—source data 1*). A combination of increased Pol II PAR-CLIP signal downstream of the CUTs following Nrd1 depletion (*Schaughency et al., 2014*) and low divergent non-coding transcription of the downstream gene in NET-seq data (*Marquardt et al., 2014*) identified *CUT95*, *CUT277*, *CUT48* and *CUT217* that shared many characteristics with *CUT60* (*Figure 3—figure supplement 2*). The genomic *CUT60* sequences were seamlessly replaced with the selected CUTs (*Figure 3—figure supplement 1G*). In addition, we replaced *CUT60* with *CUT#78* resulting from

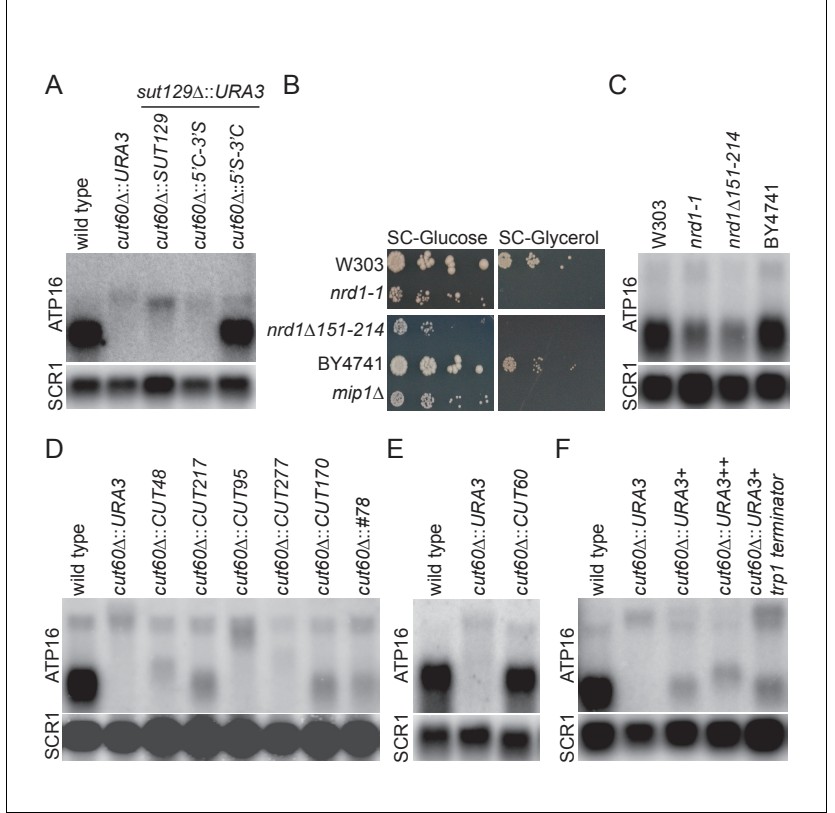

**Figure 3.** Upstream transcriptional termination promotes *ATP16* expression. (**A**) Northern blot analysis of *ATP16* and *SCR1* (as loading control) transcripts in wild type, *cut60Δ::URA3*, *cut60Δ::SUT129 + sut129Δ::URA3*, *cut60Δ::5'C-3'S + sut129Δ::URA3*, *cut60Δ::5'S-3'C + sut129Δ::URA3* strains (n = 4). (**B**) Serial dilution growth of W303 (wild type for *nrd1-1*), *nrd1-1*, *nrd1Δ151–214* and BY4741 (wild type for *nrd1Δ151–214*) strains on SC-Glucose and SC-Glycerol plates (n = 2). (**C**) Northern blot analysis of *ATP16* and *SCR1* (as loading control) transcripts in W303 (wild type for *nrd1-1*), *nrd1-1*, *nrd1Δ151–214* and BY4741 (wild type for *nrd1Δ151–214*) (n = 3). (**D**) Northern blot analysis of *ATP16* and *SCR1* (as loading control) transcripts in wild type, *cut60Δ::URA3*, *cut60Δ::CUT48*, *cut60Δ::CUT217*, *cut60Δ::CUT95*, *cut60Δ::CUT277*, *cut60Δ::CUT170* and *cut60Δ::#78* strains (n = 3). (**E**) Northern blot analysis of *ATP16* and *SCR1* (as loading control) transcripts in wild type, *cut60Δ::URA3* and *cut60Δ::CUT60* strains (n = 3). (**F**) Northern blot analysis of *ATP16* and *SCR1* (as loading control) transcripts in wild type, *cut60Δ::URA3+*, *cut60Δ::URA3++* and *cut60Δ::URA3+trp1 terminator* strains (n = 3).

DOI: https://doi.org/10.7554/eLife.31989.007

The following source data and figure supplements are available for figure 3:

**Source data 1.** Table of 68 CUTs with similar genomic configuration to *CUT60*.
DOI: https://doi.org/10.7554/eLife.31989.011

**Figure supplement 1.** Genomic replacement of *CUT60*.
DOI: https://doi.org/10.7554/eLife.31989.008

**Figure supplement 2.** Genomic configuration of CUTs selected for *CUT60* replacement.
DOI: https://doi.org/10.7554/eLife.31989.009

**Figure supplement 3.** Features preventing Transcriptional Interference highlighted in *CUT60* and CUTs selected for genomic replacements.
DOI: https://doi.org/10.7554/eLife.31989.010

synthetic selection for strong NNS termination (*Porrua et al., 2012*) and *CUT170* that lacks a downstream gene within our limits (*Figure 3—figure supplement 2*). *ATP16* expression increased compared to *cut60Δ::URA3* controls in the *cut60Δ::CUT217*, *cut60Δ::CUT170* and *cut60Δ::#78* even though not fully to wild type level (*Figure 3D*). Our selection for similarity to *CUT60* identified *CUT217* that results in relatively high *ATP16* expression in *cut60Δ::CUT217*. However, the *cut60Δ:: CUT95*, *cut60Δ::CUT277* and *cut60Δ::CUT48* replacements result in no detectable *ATP16* expression

increase (*Figure 3D*). To test if these results could be explained by differences in termination efficiency between the inserted CUTs, we quantified known binding sites for the Nab3 and Nrd1 RNA binding protein components of NNS (*Carroll et al., 2004*; *Jamonnak et al., 2011*; *Porrua et al., 2012*; *van Nues et al., 2017*; *Wlotzka et al., 2011*) (*Figure 3—figure supplement 3A–B*). The results suggest that CUTs increasing *ATP16* expression are enriched for NNS-targeting sites, even though the high number of sites for *CUT277* illustrates that this analysis is not fully predictive. We analyzed Nrd1 and Nab3 PAR-CLIP data to examine NNS-targeting to the tested CUTs in vivo (*Figure 3—figure supplement 3C–D*) (*Schulz et al., 2013*). We find high Nrd1 and Nab3 PAR-CLIP signal at *CUT60* consistent with efficient NNS termination and sensitivity to reduced NNS function. Interestingly, *CUT277* shows relatively high NNS occupancy by PAR-CLIP yet does not terminate efficiently enough to promote *ATP16* expression in the *cut60Δ::CUT277* replacement. To rule out epigenetic effects resulting from homologous recombination we reintroduced *CUT60* following *cut60Δ::URA3* replacement, essentially recreating a wild type yeast strain that has undergone the procedure of site-specific recombination (*cut60Δ::CUT60*) (*Figure 3—figure supplement 1G*). Reintroducing *CUT60* restores *ATP16* expression, arguing against epigenetic effects preventing *ATP16* expression in CUT replacement strains (*Figure 3E*). *CUT60* function can be partially provided by other lncRNA defined as CUTs, however *ATP16* expression is highest when *CUT60* sequences trigger termination. We consider a combination of strong *CUT60* termination efficiency and locus-specific effects relaying upstream termination to the activation of *ATP16* expression as the most parsimonious explanation for these results. All in all, our site-specific CUT replacement data support that NNS termination upstream of *ATP16* promotes functional *ATP16* expression.

To directly test if transcription termination upstream of *ATP16* is required for functional *ATP16* expression, we replaced *CUT60* with *URA3* including three sequences mediating efficient termination of Pol II transcription downstream of protein coding genes*Figure 3—figure supplement 1E*; *Figure 3—figure supplement 1F*; *Figure 3—figure supplement 1G*. We compared *ATP16* expression in wild type, *cut60Δ::URA3*, *cut60Δ::URA3+*, *cut60Δ::URA3++* and *cut60Δ::URA3-trp1-terminator* (*Figure 3F*). Strikingly, *ATP16* expression increases with all three strategies to enhance termination of URA3 transcription. These results strongly argue for a model in which the process of transcriptional termination promotes *ATP16* expression.

## *CUT60* promotes *ATP16* expression by preventing Transcriptional Interference

Our data are consistent with a *cis*-acting model for *CUT60* function that emphasizes the role of efficient transcriptional termination. To understand how *CUT60* promotes *ATP16* expression, we tested if chromatin signatures at the *ATP16* promoter are affected by read-through transcription. We performed Chromatin Immuno Precipitation (ChIP) coupled to quantitative PCR (qPCR) using indicated primer pairs and strains (*Figure 4A*). Read-through transcription can repress downstream gene expression by Transcriptional Interference (TI) (*Proudfoot, 1986*). If *ATP16* was repressed by TI, we expect chromatin signatures associated with Pol II elongation at the *ATP16* promoter in *cut60Δ::URA3* mutants compared to *cut60Δ::URA3++* mutants that terminate transcription or wild type controls. Promoter regions are characterized by a nucleosome depleted region (NDR) where nucleosome occupancy can be elevated by TI (*Hainer et al., 2011*). To test if the *ATP16* promoter NDR is affected by upstream termination, we performed ChIP experiments with an antibody against histone 3 (H3) and a tri-methylated version of H3 lysine 36 (H3K36me3) (*Figure 4—figure supplement 1A–C*). H3K36me3 marks Pol II elongation zones in gene bodies and performs an important function to repress cryptic initiation from within transcription units (*Carrozza et al., 2005*; *Keogh et al., 2005*; *Venkatesh et al., 2012*). The levels of H3K36me3/H3 are unchanged between wild type, *cut60Δ::URA3++* and *cut60Δ::URA3* in the gene-body region. However, H3K36me3/H3 levels are significantly increased in *cut60Δ::URA3* compared to controls in the *ATP16* promoter region (*Figure 4A*). Our ChIP analyses indicate that *ATP16* repression is associated with elevated chromatin signatures of Pol II elongation at the *ATP16* promoter. These data are consistent with regulation by TI if termination of upstream transcription is inefficient.

Previous characterizations of TI suggest that this mechanism relies on efficient Pol II elongation and associated chromatin signatures at promoter regions (*Ard and Allshire, 2016*; *Hainer et al., 2011*). If *ATP16* was indeed sensitive to repression by TI, we would expect increased expression from the *ATP16* promoter when the efficiency of Pol II elongation is reduced. As the loss of mtDNA

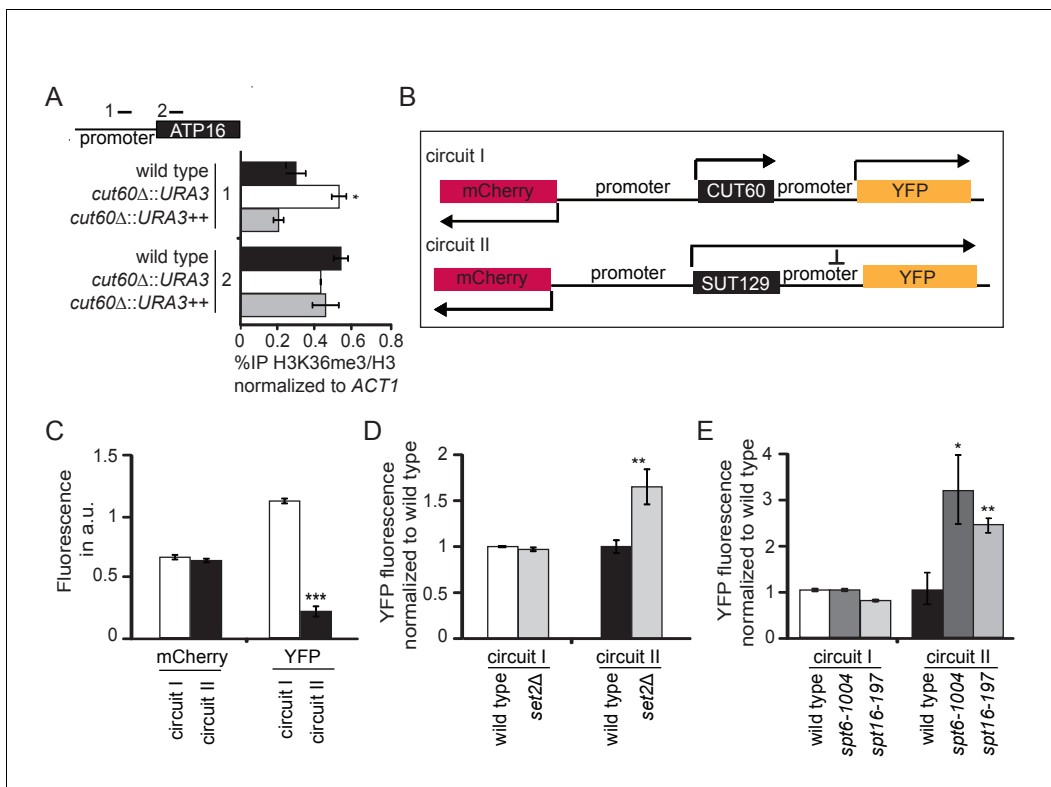

**Figure 4.** *CUT60* prevents *ATP16* repression by Transcriptional Interference. (**A**) ChIP-qPCR experiments for histone H3K36 trimethylation (H3K36me3) relative to the histone H3 levels at the promoter and gene body of *ATP16* in wild type (black), *cut60Δ::URA3* (white) and *cut60Δ::URA3++* (grey) strains. Error bars are s.e.m. of three biological replicates. * indicates p values of p<0.05 comparing %IP of H3K36/H3 in *cut60Δ::URA3* to the %IP of H3K36me3/H3 in wild type. Position of ChIP-qPCR probes indicated in top panel. (**B**) Schematic representation of synthetic circuits. *mCherry* replaces *MED2*, *YFP* replaces *ATP16*. Circuit I represents wild type circuit containing *CUT60*, circuit II represents the *cut60Δ::SUT129* mutant circuit. (**C**) Relative fluorescence of mCherry and YFP in circuit I (white) and circuit II (black). Error bars are s.e.m. of nine biological replicates, the mean was derived from multiple thousand individual measurements. *** indicates p values of p<0.0001. (**D**) Relative YFP fluorescence of circuit I in wild type (white) and *set2Δ* (light grey) background, and of circuit II in wild type (black) and *set2Δ* (light grey) background. YFP fluorescence levels are normalized to the corresponding wild-type values. Error bars are s. e.m. of six biological replicates. ** indicates p values p < 0.01 comparing YFP fluorescence in *set2Δ* to the respective wild type. (**E**) Relative YFP fluorescence in circuit I in wild type (white), *spt6*-1004 (dark grey) and *spt16*-197 (light grey) background, and in circuit II in wild type (black), *spt6*-1004 (dark grey) and *spt16*-197 (light grey) background. YFP fluorescence is normalized to the corresponding wild-type values. Error bars are s.e.m. of six biological replicates. * and ** indicate p values of p<0.05 and p<0.01 respectively comparing YFP fluorescence in *spt6*-1004 and *spt16*-194 to the respective wild type.

DOI: https://doi.org/10.7554/eLife.31989.012

The following figure supplement is available for figure 4:

**Figure supplement 1.** Set2p, Spt6p and Spt16p participate in Transcriptional Interference of *ATP16*.
DOI: https://doi.org/10.7554/eLife.31989.013

precludes growth assays to monitor functional *ATP16* expression we designed synthetic circuits to quantitatively assay expression from the *ATP16* promoter using fluorescent proteins (**Figure 4B**). We replaced the *MED2* coding sequence with mCherry and the *ATP16* coding sequence with YFP by cloning. We used the wild type DNA region containing *CUT60* (circuit I) where we expect efficient transcriptional termination upstream of the *ATP16* promoter. Consistently, we detected high levels of mCherry as well as YFP fluorescence (**Figure 4C**). We generated a second circuit based on *cut60Δ::SUT129* (circuit II) where we expect low termination efficiency upstream of the *ATP16* promoter (**Figure 3A**). YFP expression from the *ATP16* promoter in circuit II is significantly reduced compared to circuit I, while mCherry expression appears largely unaffected (**Figure 4C**). RNA

expression analysis in *cut60Δ::SUT129* represents a population average measurement. The reduced *ATP16* expression could be explained by a gradual reduction in a homogenous population, or a higher proportion of cells lacking *ATP16* expression in a heterogeneous population. YFP fluorescence is normally distributed across thousands of individual cells, supporting the hypothesis that *ATP16* expression is gradually reduced in a homogenous population (*Figure 4—figure supplement 1D*). All in all, the repression of the *ATP16* promoter by read-through transcription is supported by reduced YFP fluorescence in circuit II.

To test the role of candidate factors facilitating Pol II elongation in TI-repression of the *ATP16* promoter we combined elongation factor mutants with our synthetic circuits. We expect mutations in elongation factors required for TI to elevate YFP expression in circuit II. Since we observed elevated H3K36me3/H3 levels at the *ATP16* promoter, we tested the involvement of Set2p mediating H3K36me3 in budding yeast (*Carrozza et al., 2005*; *Keogh et al., 2005*). While mCherry expression is unchanged in *set2Δ* mutants, YFP expression increases (*Figure 4D*). This effect of *set2Δ* is not observed in circuit I (*Figure 4—figure supplement 1E*). Gene repression by TI relies on the activity of chromatin remodeling by the Spt6p and Spt16p histone chaperones (*Hainer et al., 2011*; *Kaplan et al., 2003*). Hence, we assayed fluorescence resulting from the synthetic circuits in *spt16-197* and *spt6-1004* mutants (*Kaplan et al., 2003*; *Prelich and Winston, 1993*). Both mutants increase YFP expression specifically in circuit II, consistent with TI suppression (*Figure 4E*). While the changes in fluorescence in *set2Δ* are specific to YFP fluorescence, *spt6-1004* and *spt16-197* reduce mCherry fluorescence in circuit I and circuit II (*Figure 4—figure supplement 1F*). These data show that Pol II elongation factors previously associated with gene repression by TI are required for full repression of the *ATP16* promoter region when the termination of upstream lncRNA transcription is inefficient. These results support the notion that efficient *CUT60* termination prevents *ATP16* repression by a TI mechanism.

In conclusion, our data suggest that efficient termination of upstream lncRNA transcription ensures *ATP16* promoter activity. *CUT60* termination promotes functional *ATP16* expression, the maintenance of mitochondrial DNA and yeast growth. All in all, the dependence of mitochondrial DNA stability on *CUT60* termination provides a compelling example for the biological significance of pervasive non-coding transcription in genomes.

## Discussion

### CUT60 termination promotes mitochondrial genome maintenance

While pervasive transcription of Pol II in eukaryotic genomes results in the production of many lncRNA, their functional significance often remains elusive. Our analysis of the lncRNA *CUT60* reveals how efficient transcriptional termination fulfills an important biological role. Inefficient transcriptional termination of *CUT60* causes read-through transcription across the promoter of the downstream *ATP16* gene, which represses *ATP16* transcription (*Figure 5*). *CUT60* read-through transcription represses functional *ATP16* expression by a Transcriptional Interference mechanism. *ATP16* repression results in a striking biological defect: mitochondrial genome loss ($\rho^0$). The function of Atp16p as δ-subunit of the mitochondrial ATP-synthase can reconcile the strong phenotypic response resulting from inefficient *CUT60* termination. Interestingly, in *atp16* mutant backgrounds the loss of the mitochondrial genome represents a survival strategy (*Duvezin-Caubet et al., 2006*). At first glance the positive selection for the loss of the mitochondrial genome in *atp16* mutants seems counter-intuitive. However, acquiring $\rho^0$ efficiently blocks proton leakage, as the two subunits encoding the core of the ATP-synthase proton channel (*ATP6* and *ATP9*) are located on the mitochondrial genome. Hence, loss of the mitochondrial genome disrupts proton channel formation to prevent uncoupling of the ATP synthase in the absence of a functional δ-subunit (*Duvezin-Caubet et al., 2006*). Future research will be required to identify if modulating *CUT60* termination efficiency is used as mechanism for regulated mtDNA loss. Mutant mtDNA can out-compete wild type mtDNA copies through increased replication efficiency (*Contamine and Picard, 2000*). It is tempting to speculate that the loss of mtDNA triggered by transiently reduced *CUT60* termination efficiency could be a temporary state to detoxify cells from deleterious mtDNA. Healthy mtDNA variants could be subsequently acquired through mating and would be maintained as long as *CUT60* transcription is terminated efficiently. In human, mutant mtDNA has emerged as common cause for metabolic disease and

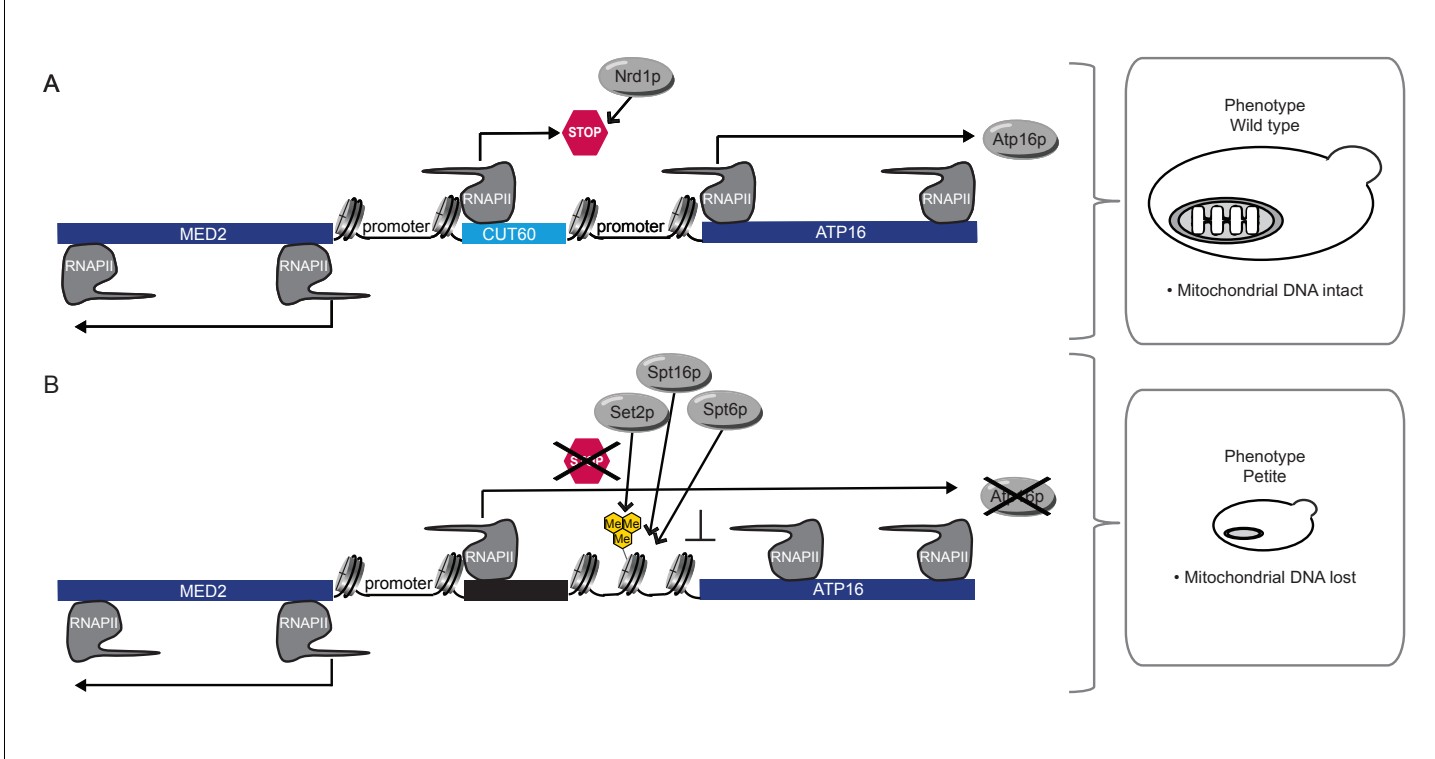

**Figure 5.** Model for *CUT60* function as insulator from Transcriptional Interference. (**A**) In the presence of *CUT60* transcription initiating from the bidirectional *MED2* promoter terminates efficiently with the help of Nrd1p to prevent Transcriptional Interference of *ATP16*. Atp16p promotes the maintenance of mitochondrial DNA. (**B**) *ATP16* is repressed by Transcriptional Interference if *CUT60* is replaced with sequences terminating inefficiently. Interfering transcripts initiate from the bidirectional *MED2* promoter. The interference mechanism is enforced by Set2p, Spt6p and Spt16p. This leads to the loss of mitochondrial DNA triggered by the lack of Atp16p.

DOI: https://doi.org/10.7554/eLife.31989.014

mechanisms to acquire healthy mtDNA would offer a promising avenue for biotechnology (*DiMauro et al., 2013*). All in all, our findings provide a compelling example for a biologically significant lncRNA transcription termination event. Promoting the stability of organellar DNA adds a new function to the growing list of 'junk DNA'-derived lncRNA with functional roles in cells.

## Gene regulation by divergent lncRNA transcription

Divergent transcription of lncRNA from eukaryotic gene promoters represents an important source of lncRNA (*Sigova et al., 2013*). Divergent lncRNA transcription is repressed by lncRNA degradation (*Almada et al., 2013*; *Ntini et al., 2013*; *Xu et al., 2009*), and by a repressive chromatin architecture (*Jin et al., 2017*; *Marquardt et al., 2014*; *Rege et al., 2015*; *Tan-Wong et al., 2012*). Transient inhibition of these pathways and a reduction of termination efficiency could amplify read-through transcription of divergent lncRNA (*Mellor et al., 2016*). Inefficient transcriptional termination and environmentally regulated expression define the species of budding yeast SUTs (*Marquardt et al., 2011*; *Xu et al., 2009*). These characteristics poise SUTs for gene regulation by read-through transcription, for example at *IME1* and *GAL80* (*van Werven et al., 2012*; *Xu et al., 2011*). The TI mechanism described in our study is reminiscent of the regulation of *FLO11* expression and yeast flocculation by a divergently transcribed SUT lncRNA (*Bumgarner et al., 2009*). While regulation of *FLO11* expression is regulated at the level of individual cells resulting in heterogeneous expression states (*Bumgarner et al., 2012*), our analysis of circuit II suggests that *ATP16* expression is reduced in a homogenous population. Replacement of the efficiently terminated *CUT60* with inefficiently terminated *SUT129* supports a role for SUTs as source for read-through regulation. The 5′-SUT129-3′CUT60 hybrid experiments in particular suggest that the underlying molecular cause may be differences in termination efficiencies of CUTs and SUTs (*Figure 3*). While the SUT nomenclature is

specific to budding yeast, it should be noted that many human lncRNA share characteristic cell-type-specific and environmentally regulated expression (*Deveson et al., 2017*) and *cis*-acting functions are increasingly appreciated (*Engreitz et al., 2016*). Divergent lncRNA transcription of human promoter upstream transcripts (PROMPTs) may overlap with downstream genes reminiscent of the *MED2-CUT60-ATP16* circuitry (*Chen et al., 2016*). Efficient transcriptional termination of a divergent lncRNA has therefore functional implications beyond the specific example described here.

## Functional diversification within lncRNA subsets

The sensitivity to RNA degradation pathways has been used as a defining criterion to distinguish lncRNA species in budding yeast (*van Dijk et al., 2011*; *Xu et al., 2009*). It remains unclear how frequently lncRNA classification based on RNA decay pathways is indicative of equivalent lncRNA function. Our study addresses this question by testing experimentally which lncRNAs of the CUT species can functionally substitute for *CUT60* (*Figure 3*). To enable these experiments we systematically identified CUTs with similar genomic location to examine if they can replace *CUT60* function (*Figure 3—figure supplement 2*). Indeed, *ATP16* expression increases in a subset of CUTs, but none of the CUTs we tested fully restores *ATP16* expression. Our results argue that there is some yet limited functional overlap between lncRNA classified as members of the same species. Importantly, our analysis of complementation plasmids shows that sequences downstream of *CUT60* are sufficient to fully restore *ATP16* expression (*Figure 2*). DNA sequence elements in *CUT60* that could hypothetically provide promoter function can therefore not explain reduced *ATP16* expression in the replacement mutants. We detect strong binding of Nrd1 and Nab3 to *CUT60* by PAR-CLIP, suggesting *CUT60* represents a strong transcriptional terminator for the NNS pathway (*Figure 3—figure supplement 3*). Consistently, we find that hypomorphic mutations in Nrd1 result in reduced *ATP16* expression, *CUT60* read-through transcription and common growth phenotypes with *CUT60* termination defective strains (*Figures 1–3*). While *CUT60* scores highly in accessible metrics to predict NNS termination these differences cannot fully account for the termination efficiency and resulting *ATP16* expression, particularly in comparison to *CUT277* (*Figure 3—figure supplements 2,3*). We map sequences mediating strong termination of Pol II transcription to the 3'-half of *CUT60* where a cluster of NNS-targeting *cis*-elements is located (*Figure 3—figure supplement 3*). Moreover, we can partially restore *ATP16* expression by the inclusion of three sequences to trigger transcriptional termination by NNS-independent pathways (*Figure 3F*). The most parsimonious interpretation of our experimental results is that an architecture linking efficient transcriptional termination of *CUT60* with the initiation of *ATP16* expression is under strong locus-specific selection, presumably to maintain the mitochondrial genome.

## Insulating the *ATP16* promoter from interfering upstream transcription

Even though most promoters in eukaryotic genomes initiate transcription bi-directionally (*Preker et al., 2008*; *Seila et al., 2008*), the *ATP16* promoter shows little evidence of divergent lncRNA transcription (*Figure 1—figure supplement 1A*, *Figure 3—figure supplement 2A*). Low levels of divergent lncRNA transcription may offer a partial explanation for the need of *CUT60* as strong 'insulator' for the *ATP16* promoter. Divergent lncRNA transcription could conceivably provide an opposing wave of Pol II transcription that may trigger termination of incoming read-through transcripts by polymerase collisions, as proposed for the yeast *BAT2* promoter (*Hobson et al., 2012*; *Mellor et al., 2016*; *Nguyen et al., 2014*). The extended *ATP16* transcript we detect in the background of construct V1 lacking *CUT60* may be a manifestation of this phenomenon (*Figure 1D*). The extended transcript likely results from upstream transcription initiation in the plasmid, as it is not observed in constructs containing *CUT60* sequences that would lead to termination. However, our site-specific swaps of CUTs do not support that lncRNAs upstream of unidirectional gene promoters are generally characterized by particular efficient NNS termination (*Figure 3*). The *ATP16* promoter lacks binding sites for the Reb1p and Rap1p transcription factors that prevent read-through transcription by a road-blocking mechanism, which may also contribute to the sensitivity for read-through repression (*Candelli et al., 2018*). It has previously been established that promoter repression by TI can be partially overcome by increasing the levels of promoter-specific transcription factors to trigger transcriptional initiation (*Greger et al., 2000*). A locus-specific balance of

transcriptional read-through across promoters and their strength to mediate transcriptional initiation likely determines the susceptibility of promoters for repression by TI.

## Gene regulation through the act of read-through transcription

Low primary sequence conservation combined with low steady-state abundance of lncRNA challenges functions of pervasive transcription through the resulting lncRNA molecules. Our experimental characterizations of *CUT60* reveal a *cis*-acting lncRNA function through the mechanism of NNS-termination to promote *ATP16* expression. We propose that *ATP16* promoter function is inhibited through a TI mechanism by upstream transcription extending across this region when *CUT60* termination is inefficient (*Figure 5*). A TI mechanism is consistent with an increase in the Pol II elongation-specific H3K36me3 mark at the *ATP16* promoter region that is usually detected in the middle of transcription units. In addition, the Set2p histone-methyl transferase mediating H3K36me3 is genetically required for efficient repression of the *ATP16* promoter (*Figure 4*). Set2p is required for TI by sufficiently long upstream interfering transcripts (*Kim et al., 2016*). The histone chaperones Spt6p and Spt16p are required for TI without an apparent size limit of the interfering transcript (*Ard and Allshire, 2016*; *Hainer et al., 2011*; *Winston et al., 1984*). We find that both histone chaperones are also required to enforce TI repression of the *ATP16* promoter. While gene regulation by TI may be most intuitive in dense genomes, this mechanism has been first described for two tandem copies of the human α-globin genes (*Proudfoot, 1986*). Human lncRNA sequences are rich in cis-elements triggering polyadenylation and could help to protect from TI akin to *CUT60* (*Almada et al., 2013*; *Ntini et al., 2013*). Read-through transcription in human is associated with cellular responses to viral infection (*Rutkowski et al., 2015*) and osmotic stress (*Vilborg et al., 2015*). Future research elucidating the molecular mechanism of gene repression through the act of transcription will be required to detect and predict where this powerful mechanism operates in genomes. While previous studies largely focused on lncRNA transcription as a cause for TI (*Ard et al., 2017*), our results suggest that lncRNA transcription may also offer protection. In summary, our findings expand the functional and mechanistic implications of 'junk DNA' transcription in eukaryotes.

## Materials and methods

**Key resources table**

| Reagent type (species) or resource | Designation | Source or reference | Identifiers | Additional information |
|---|---|---|---|---|
| strain, strain background (*S. cerevisiae*) | Yeast Strains | in this study | | See *Supplementary file 1* Yeast strains |
| strain, strain background (*E.coli*) | E. coli | Thermo Fisher | Cat#18265017 | E. coli DH5α Competent Cells |
| antibody | H3 | Abcam | Ab1791; RRID: AB_302613 | Anti-Histone H3 |
| antibody | H3K36me3 | Abcam | Ab9050; RRID: AB_306966 | Anti-Histone H3 tri methyl K36 |
| commercial assay or kit | QIAprep Spin Miniprep Kit | QIAGEN | Cat#27106 | |
| commercial assay or kit | Wizard SV Gel and PCR Clean-Up System | Promega | Cat#A9182 | |
| commercial assay or kit | MasterPure Yeast RNA Purification Kit | Lucigen | MPY03100 | |
| chemical compound, drug | Protein A magnetic beads | GenScript | Cat. No. L00273 | |
| chemical compound, drug | Phusion high-fidelity PCR master mix | NEB | cat. no. M0531S | |
| chemical compound, drug | GoTaq qPCR Master Mix | Promega | Cat#A6002 | |
| chemical compound, drug | NAT | Jena Bioscience | AB-102L | Nourseothricin dihydrogen, cloNAT |
| chemical compound, drug | G418 | Duchefa-biochemie | G0175 | G-418 disulphate |
| chemical compound, drug | 5FOA | Thermo Fisher | R0811 | 5-Fluoroorotic Acid |
| software, algorithm | ImageQuant | GE Healthcare | NA | |

*Continued on next page*

*Continued*

| Reagent type (species) or resource | Designation | Source or reference | Identifiers | Additional information |
|---|---|---|---|---|
| software, algorithm | Software for flow cytometry and bioinformatic analysis: see Materials and methods | NA | NA | |
| other | Oligonucleotides | in this study | NA | See *Supplementary file 2* Oligonucleotides |
| other | Equipment: BD Fortessa | BD biosciences | NA | |
| other | Plasmids | in this study | NA | See *Supplementary file 3* Plasmids |

## Strains

Yeast strains used in this study are listed in *Supplementary file 1*. Strains were constructed using standard procedure.

To introduce *cut60Δ::URA3* mutation primers SB2803 and SB2804 were used with SMC88 as template DNA. Primers SB2803 and SB2804 amplify *URA3* coding region and contain overhangs that are homologous to the region upstream and downstream of the annotated *CUT60* sequence (*Xu et al., 2009*). These homologous overhangs integrate PCR fragments into the desired locus as described in *Figure 3—figure supplement 1G*. Transformants were selected using SC-URA plates. To generate *cut60Δ::URA3#1*, *cut60Δ::URA3#2* and *cut60Δ::URA3#3* primers MLO515 and MLO516 were used on genomic DNA using SMY81 as template. The PCR fragments containing the *cut60Δ:: URA3* replacement and homologous overhangs to the region upstream and downstream of *CUT60* were transformed into SMY2132 and selected for growth on SC-URA plates. To generate *cut60Δ:: CUT60#1* and *cut60Δ::CUT60#1* primers MLO488 and MLO489 were used SMY2132 genomic DNA template. The generated PCR fragments containing the endogenous *CUT60* sequence and homologous overhangs to the region upstream and downstream of *CUT60* and were transformed into SMY81 and selected using 5-FOA plates. *cut60Δ::SUT129* mutants were generated starting from *cut60Δ::URA3* mutant strains. *URA3* was replaced with a PCR fragment from genomic DNA using primers SB2875 and SB2876. These primers contain homologous overhangs to the region upstream and downstream of *CUT60*. Transformants were selected using 5-FOA plates. To generate *cut60Δ::5'C-3'S* mutants, we generated PCR fragments using primers SB2823 and SB2899 on genomic *CUT60* DNA as template DNA, and primers SB2898 and SB2876 on genomic *SUT129* on genomic DNA as template. These fragments were fused by PCR using primers SB2823 and SB2876, and these fusion fragments were then transformed into *cut60Δ::URA3* strains. Transformants were selected using 5-FOA plates. To generate *cut60Δ::5'S-3'C* mutants, we generated PCR fragments using primers SB2900 and SB2902 on genomic *CUT60* template DNA and primers SB2875 and SB2901 on genomic *SUT129* template DNA. These fragments were fused by PCR using primers SB2875 and SB2902, and the resulting fusion fragment was transformed into *cut60Δ::URA3* strains. Transformants were selected using 5-FOA plates. To introduce *rrp6Δ::KanMX* mutations, we generated PCR fragments for transformation using primers SB1055 and SB1056 on *rrp6Δ::KanMX* template DNA. Transformants were selected on G418 plates. To introduce *xrn1Δ::KanMX* mutations, we generated PCR fragments for transformation using primers SB2877 and SB2878 on *xrn1Δ::KanMX* template DNA. Transformants were selected on G418 plates. To generate strains where *CUT60* was replaced with different lncRNA sequences, we transformed *cut60Δ::URA3* strains with PCR fragments that contains the lncRNA sequence of interest and homologous overhangs to the region upstream and downstream of *CUT60* as in *Figure 3—figure supplement 1G*. Primers MLO1319 and MLO1320 were used for *cut60Δ::CUT170*, primers MLO496 and MLO497 were used for *cut60Δ:: CUT95*, primers MLO498 and MLO499 were used for *cut60Δ::CUT277*, MLO1376 and MLO1377 were used for *cut60Δ::CUT48*, MLO1370 and MLO1371 were used for *cut60Δ::CUT217*, MLO1328 and MLO1329 were used for *cut60Δ::CUT#78*. Transformants were selected using 5-FOA plates. *cut60Δ::URA3 ++* contains the coding sequence of *URA3* and the 300 bp downstream of the *URA3* coding sequence on pRS316 as terminator. Primers MLO933 and MLO935 are used on pRS316 template DNA to generate a PCR fragment containing the *URA3* coding sequence including terminator and homologous overhangs to the sequence upstream and downstream of *CUT60*. Transformants

were selected using SC-URA plates. *cut60Δ::URA3+* contains the coding sequence of *URA3* and the 80 bp downstream of the endogenous coding sequence as terminator. Primers MLO933 and MLO934 are used on pRS316 template DNA to generate a PCR fragment containing the *URA3* coding sequence including the 80 bp terminator with homologous overhangs sequences upstream and downstream of *CUT60*. Transformants were selected using SC-URA plates.

*ut60Δ::URA3 +trp1 terminator* contain the coding sequence of *URA3* and the 80 bp downstream of the coding sequence of the endogenous *TRP1* gene as terminator. Primers MLO933 and MLO1368 are used on pRS316 template DNA to generate a PCR fragment containing the *URA3* coding sequence with homologous overhangs to the sequence upstream of *CUT60* and the *TRP1* terminator 5' end. MLO1335 and MLO1334 are used on SMC147 template to generate a PCR fragment containing the 80 bp *TRP1* terminator that has homologous overhangs to the 3' sequence of URA3 and the sequence downstream of *CUT60*. Transformants were selected using SC-URA plates.

## Growth media

Standard yeast media were used, 100 µg/ml ampicillin was added to liquid media to avoid bacterial contaminations. YPD liquid media and plates containing 300 µg/ml G418 Sulfate (Geniticin, American Bioanalytical) were used to select for KanMX or/and 100 µg/ml ClonNat (Jena Bioscience, Jena, Germay) to select for NatMX. SC media lacking leucine was used for growth of yeasts containing the complementation plasmids (V0-V4). SC media containing uracil was used for selection for growth in presence of 1 mg/ml 5-Fluoro Orotic Acid (5-FOA), to select against cells with functional *URA3*. SC-Glycerol and SC-Ethanol plates are used to test for mitochondrial function.

## Transformations

Single colonies of yeast strains were incubated in 5 ml of liquid medium (YPD or SC-Leu medium) overnight in a shaker at 150 RPM at 30°C. The titer was determined, using spectrophotometer. Cultures were diluted and harvested at a titer of $2 \times 10^7$ cells/ml. Per transformation reaction, 1/10 of the harvested pellet was used. Transformation mix (240 µl PEG (50% w/v), 36 µl 1M LiAc, 10 µl single stranded salmon sperm carrier DNA (10 mg/ml), 74 µl PCR product or 5–10 µl of plasmid, 34 µl $H_2O$ minus the volume of PCR product or plasmid was added to the cell pellets and mixed by pipetting. Cells were then heat shocked at 42°C for 30 min. For selection using *URA3* or *LEU* markers cells were plated on SC-URA or SC-LEU plates directly. Selection for G418, NAT or 5-FOA resistance was performed following growth on YPD plates for 1 day and then replica-plated to an appropriate selection plate. Transformants were mostly visible after 3 days.

## Plasmid construction

SMC340, SMC342, SMC361 and SMC362 were constructed using one step isothermal assembly of four overlapping dsDNA fragments as described (*Gibson, 2011*). Fragments that were inserted into HindIII digested SMC339 backbones, were generated by PCR reaction using primers MLO431, MLO433, MLO486, MLO487 as forward primers and MLO434 as reverse primer. Vectors were sequenced with primers MLO434, MLO486, MLO487, MLO489 and MLO467.

SMC425 and SMC426 were constructed using one step isothermal assembly. The fragments that were inserted into AscI digested SMC50 backbones were generated by PCR reactions. The mCherry-NatMX fragment was made using primers MLO517 and MLO777; YFP fragment was made using primers MLO517 and MLO778 both with SMC50 as template. The central fragment containing the bidirectional promoter of *MED2* and either the *CUT60* sequence (for circuit I) or the *SUT129* sequence (for circuit II) and were amplified using primers MLO776 and MLO779, using BY4741 as template for the *CUT60* version and SMY131 (*cut60Δ::SUT129*) as template for the *SUT129* version. The three fragments were fused using PCR with splicing overlapping ends, using each other and more of the flanking primer MLO517 to make one continuous fragment, which was identified on an agarose gel. Plasmids were genotyped by PCR and sequenced.

## Serial dilution growth assay

Overnight cultures were grown at 30°C in media. The yeast pellets were washed with water, and diluted to $OD_{600} = 0.1$. An equal starting amount of cells were spotted in three-fold serial dilutions and growth assayed at 30°C for 2–3 days.

## RNA isolation

Single colonies of yeast strains were incubated in 5 ml of liquid medium overnight in a shaker at 150 RPM at 30°C. Subsequently 25 ml of liquid medium were inoculated to an $OD_{600}$ = 0.1 and grown at 30°C to $OD_{600}$ = 0.4–0.8. RNA was isolated using the MasterPureTM Yeast RNA Purification Kit (Epicentre, Wisconsin, USA). Final RNA concentration was determined using a Nanodrop spectrophotometer.

## Northern blots

Ten micrograms of total RNA was separated by electrophoresis on 1.5% agarose-formaldehyde-MOPS gels. The RNA was transferred to a nylon transfer membrane via capillary blotting in 10 X SSC buffer for 16 hr. RNA was cross-linked to the nylon membrane by UV irradiation. The membrane was incubated in a rotating oven at 65°C in 15 ml hybridization buffer (0.5 g BSA, 15 ml phosphate buffer, 35 ml 10% SDS, 100 µl 0.5 M EDTA pH 8.0; 100 ml phosphate buffer consisting of 68.4 ml 1M $Na_2HPO_4$68.4 ml and 31.6 ml 1 M $Na_2H2PO_4$*$H_2O$) 60 min prior to addition of radioactive probe. Single-stranded probes were generated by incorporation of radioactive dTTP into DNA using a unidirectional thermocycling reaction. Twenty microliters of reaction mix were assembled containing five units Taq DNA polymerase, 200 µM each of dCTP, dGTP and dATP, 5 ng of DNA template consisting of a 100–300 bp long PCR product of the probed region, 0.4 µM primer oligonucleotide (antisense to the RNA to be detected) and 4 µl of $\alpha-32$P-dTTP added. The tube was placed into a PCR machine and subjected to 35 cycles of the following series: denaturation at 94°C for 30 s, annealing at an experimentally determined temperature for 20 s, and extension for 45 s at 72°C. Upon completion, 20 µl of water was added to the reaction and unincorporated nucleotides were removed using a Spin-50 gel-filtration column (BiomaxInc., Planegg, Germany) according to manufacturers' instructions. The purified probe was denatured by incubation at 95°C for 5 min and chilled on ice. The probe was added to the membrane in 15 ml hybridization solution and incubated overnight at 65°C in a rotating oven. The membranes were washed with low stringency wash buffer (0.1 x SSC, 0.1% SDS) and when necessary to reduce background, high stringency wash buffer (2 x SSC, 0.1% SDS) at 65°C. A Typhoon scanner and ImageQuant software (GE Healthcare Chicago, Illinois) was used for analysis and quantification. *ATP16* probe template was amplified using primers MLO466 and MLO467. *COX3* probe template was amplified using primers MLO843 and MLO844. *URA3* probe template was amplified using primers SB1777 and SB1778. *MED2* probe template was amplified using primers SB3080 and SB3081. *ATP16*-promoter probe template was amplified using primers SB2451 and SB2452. *SUT129* probe template was amplified using primers SB2453 and SB2454. The blots were stripped using 0.1% SDS and re-probed with *SCR1* loading controls. *SCR1* probes generated using MLO689 and MLO690.

## Chromatin immunoprecipitation (ChIP)

Single colonies of yeast strains were incubated in 25 ml of YPD overnight in a shaker at 150 RPM at 30°C. 250 ml YDP cultures were inoculated with overnight culture to $OD_{600}$ = 0.1 and grown until $OD_{600}$ = 0.6. Cells were fixed with 1% Formaldehyde for 15 min at room temperature. 37.5 ml of 3M Glycine was added and incubated for 5 min. Fixed cells were centrifuged at 3.500 RPM for 2 min, washed twice with ice-cold PBS (137 mM NaCl, 2.7 mM KCl, 10 mM $Na_2HPO_4$, 1.8 mM $KH_2PO_4$, pH 7.4). Cell pellets were flash frozen in liquid nitrogen and stored at −80°C. The pellet was resuspended in 1 ml ice cold FA lysis buffer +Proteinase inhibitor (Roche, Basel, Switzerland)/0.5% SDS (2X FA buffer: 100 mM Hepes-KOH pH7.5, 300 mM NaCl, 2% Triton-X-100, 0.2% Na-Deoxycholate, 2 mM EDTA) and lysed by bead beating. Crude whole cell extract was collected by puncturing small holes in the tube using flame-headed needle and centrifugation into a new microfuge tube at 1.000 RPM for 1 min. Crude whole cell extract was briefly vortexed and then sonicated at 5°C using Q700 sonicator (Qsonica, Connecticut) on Amplitude 100 for a total of 20 min (30 s ON/OFF cycles). Insoluble material was removed by centrifugation at 13.500 RPM 2 × 10 min. 25 µL samples of whole cell extracts were collected as total input controls ('Input') and frozen at −20°C. 250 µL of soluble lysates were pre-cleared with Protein A beads (Gen Script, New Jersey) for 1 hr at 4°C and then incubated with appropriate antibody and beads overnight at 4°C. 2 µL of H3 antibody (ab1791; Abcam, Cambridge, UK), and 2 µL of H3K36me3 antibody (ab9050; Abcam) were used for IPs. IPs were washed in four steps, each wash step was performed for 4 min at 4°C. First wash step was done in 1x FA

buffer/0.1% SDS +275 mM NaCl, $2^{nd}$ wash step was done in 1x FA buffer/0.1% SDS +500 mM NaCl, $3^{rd}$ wash step was done in 10 mM TRIS-HCL (pH8) +0.25 M LiCl, 1 mM EDTA, 0,5% Na-Deoxycholate and $4^{th}$ wash was done in TE buffer. Beads following IP and 25 μL of Input samples were incubated with 100 μL of 1% Chelex100 Resin (Bio-Rad) in $dH_2O$, boiled to remove DNA-protein crosslinks for 15 min, and then treated with proteinase K (10 mg/mL) for 30 min at 55°C. Samples were boiled for an additional 10 min to denature proteinase K. Samples were centrifuged at 10.000 RPM for 1 min. 60 μL of supernatant was carefully pipetted into new microfuge tubes. Quantitative analysis was performed by qPCR on diluted samples: Input DNA samples were diluted 1/100 in $dH_2O$ while IP DNA samples were diluted 1/50.

ChIP enrichments per antibody was calculated using the following formula: 2((C$_T$ Value of input)-(C$_T$ Value of IP))*(Dilution factor input/Dilution factor IP). This gives the %IP per antibody, these values were normalized to the %IP calculated for the household gene (*ACT1*), to give the %IP relative to actin. Triplicate biological samples were each analysed using three technical repeats. For normalization to the *ACT1* household gene using the primers MLO1247 and MLO1250 were used.

## qPCR

qPCR was performed using 5 μl of GoTaq qPCR Master Mix (Promega, Wisconsin), 1 μl primers (20 μM), 3 μl $H_2O$ and 1 μl of DNA in C1000 Touch Thermal Cycler (BioRad, California). For each sample, at least three biological repeats and three technical replicates were used. The amplification efficiency of each amplicon used was determined, and only amplicons with efficiencies between 97 and 110% were retained.

## Flow cytometry

A BD Fortessa (BD biosciences, New Jersey) flow cytometer with a high-throughput resolution sampler was used to quantitate YFP and mCherry fluorescence. YFP was excited at 450 nm and fluorescence collected through a 535/45 band pass and 525 LP emission filter. mCherry was excited at 600 nm and assayed with a 632/22 band pass filter. Between 30000 and 50000 events were sampled for each well. Flow cytometry data were exported from the acquisition program (FACSDiva, Beckton Dickinson, San Jose, CA, USA) in the FCS3.0 format. R studio was used to process the data, importing using a custom modified version of 'flowCore' package from Bioconductor.org. To compare cells with the same average sizes and filter out cellular debris and aggregates, the median side scatter (SSC) was used. Data within ±25% of the median SSC value were used for all the comparisons. The first third of events and the last ninth of data in time were removed to minimize errors due to unstable sample flow through the flow cytometer. Any well that had fewer than 500 counted cells was excluded from the analysis. BY4741 without a fluorescent reporter (FPR) served as background control for experiments in *Figure 4D and E*. SMY921 without FPR served as background control for experiments in *Figure 4F*. After background subtraction, the data were represented normalized to co-assayed isogenic wild-type FPR control strains. Values for circuit I were normalized to wild type FPR control strains containing circuit I, values for circuit II were normalized to wild type FPR control strains containing circuit II.

## Statistical analysis

Statistical tests were performed on means generated from at least three biological replicates. Repeats per individual experiment shown in legend (n = number of biological replicates). Results are expressed as the mean ±standard error of the mean (s.e.m.) between experimental groups. Significance is assessed using a two-tailed unpaired Student's t-tests. The *p<0.05, **p<0.01, and ***p<0.001 levels were considered significant.

## Computational analysis

Genomic coordinates of CUTs (n = 925) and SUTs (n = 847) were obtained from (*Xu et al., 2009*). The coordinates were converted from SacCer2 to SacCer3 using kentUtils liftOver tool and the chain file sacCer2ToSacCer3.over.chain.gz. Coordinates of SGD genes in SacCer3 (n = 6692) were downloaded from the UCSC Table Browser. To obtain CUTs for *Figure 3—source data 1*, CUTs overlapping other transcription units are removed. For the remaining CUTs (n = 238), the nearest downstream gene on the same strand was found, and the distance to this gene was calculated. The

final 68 CUT-gene pairs were selected based on the following criteria: (i) the distance between the 3' end of the CUT and the TSS of the downstream gene (gap interval) is between 100 bp and 1500 bp; (ii) there are no transcription units overlapping with the gap interval on either strand. This pipeline was implemented in the 01-Candidate_CUTs.R script. The directionality of the downstream genes was obtained using yeast NET-Seq data from (*Marquardt et al., 2014*) (accession GSE55982). The raw reads were extracted from SRA archives using SRA Toolkit v2.6.3, quality and adapter trimming was performed using Trim Galore v0.4.3 and alignment to the yeast genome was done using Bowtie2 (–very-sensitive-local mode). The SAM files were filtered from unmapped reads and reads with low mapping quality (MAPQ <= 10) and then converted to sorted BAM files using Samtools v1.3.1. BAM files were filtered from reads aligning to unwanted sequences such as rRNA, tRNA, snRNA and snoRNA genes and then converted into stranded Bedgraph files using Bedtools v2.25.0. Only 5' positions of reads were used to calculate the coverage. Bedgraph files were then normalized to 1 million tags. For more details, see the 02-Reanalysis_of_NET-Seq_raw_data.sh script. At the next step, these Bedgraph files were used to compute the NET-Seq coverage around TSS of genes found downstream of our candidate CUTs (n = 68).The signal at the first 100 bp of downstream gene on the coding strand gave transcription intensity into the coding direction and the signal at the last 100 bp of the gap interval on the opposite strand gave transcription intensity into the divergent direction. To avoid possible division by zero, a pseudocount of 0.25 was added to the latter values. This pipeline is detailed in the 03-Directionality_of_downstream_genes.R script.

Pol II PAR-CLIP data from *Schaughency et al., 2014* were used to assess Nrd1-dependent transcription termination of each CUT. The level of nascent transcription over the CUTs and their gap intervals were assessed using the original Wig files provided by authors (accession GSE56435). The read-through index is defined as the ratio of Pol II PAR-CLIP signal at the gap interval divided by Pol II PAR-CLIP signal at their CUT in the Nrd1-AA sample compared the control sample. The Rbp2 strain treated with rapamycin was used as the control. For details see the 04-Nrd1-dependent_termination_of_CUTs.R pipeline. The Pol II PAR-CLIP data were also used to visualize the genome-wide changes in the nascent transcription profile which occur due to the depletion of Nrd1 in *Figure 3—figure supplement 2*. We subtracted the rapamycin-treated negative control track from the Nrd1-AA track at all genomic positions. This was done by the 07-Subtract_PolII_PAR-CLIP_tracks.R script. The resultant Bedgraph files contain both positive (black) and negative (gray) values.

Nrd1 PAR-CLIP data from (*Schulz et al., 2013*) were used to assess the Nrd1 and Nab3 occupancy at each CUT. This study provides both Nrd1 PAR-CLIP and 4tU-Seq data for the same samples (accession E-MTAB-1766). The raw NGS reads were aligned to the yeast genome using STAR v2.5.2 in the local mode. The Nrd1 PAR-CLIP BAM files were converted to the mpileup format using samtools. The cross-linking positions (defined as T-to-C and A-to-G conversion sites on the forward and reverse strands, respectively) were detected with single nucleotide resolution using VarScan v.2.4.3. The resultant VCF files were converted to Bedgraph files and normalized to 1 million tags. This pipeline is described in the 05-Reanalysis_of_4sU-Seq_and_Nrd1_PAR-CLIP_raw_data.sh. The PAR-CLIP signal was also normalized by the transcription level of the underlying locus. To this end, the 4sU-Seq coverage representing nascent transcription was smoothed using the sliding window approach. A pseudocount of 0.1 was added to genomic positions with zero 4sU-Seq coverage, then mean values were taken over 51 bp windows centered around each genomic position. The Nrd1 PAR-CLIP signal was divided by the smoothed 4tU-Seq signal, giving the normalized Nrd1 occupancy track. The 4sU-normalized Nrd1 PAR-CLIP Bedgraph files were used to quantify Nrd1 occupancy over each CUT. These calculations were implemented in the 06-Quantification_of_Nrd1_occupancy_in_CUTs.R script.

All custom scripts and pipelines for data processing were deposited at (*Ivanov, 2018*). A copy is archived at https://github.com/Maxim-Ivanov/du_Mee_et_al_2018_eLife.

## Acknowledgements

SM is supported by a Halls-Møller Investigator Award by the Novo-Nordisk Foundation NNF15OC0014202, a Copenhagen Plant Science Centre Young Investigator Starting Grant and a Feodor-Lynen Research Fellowship of the Alexander von Humboldt Foundation. Funding for the BD Fortessa flow cytometer is provided by the research infrastructure grant CF14-0464 of the Carlsberg Foundation. SB is supported by GM56663 from the US National Institutes of Health. We thank

Jasmin Dilgen for help with media preparation and members of the SM lab for feedback on the manuscript.

## Additional information

### Funding

| Funder | Grant reference number | Author |
|---|---|---|
| Novo Nordisk | NNF15OC0014202 | Dorine Jeanne Mariëtte du Mee Sebastian Marquardt |
| Alexander von Humboldt-Stiftung | | Sebastian Marquardt |
| Carlsbergfondet | CF14-0464 | Sebastian Marquardt |
| National Institutes of Health | GM56663 | Stephen Buratowski |

The funders had no role in study design, data collection and interpretation, or the decision to submit the work for publication.

### Author contributions

Dorine Jeanne Mariëtte du Mee, Conceptualization, Resources, Data curation, Supervision, Funding acquisition, Validation, Investigation, Methodology, Writing—original draft, Project administration, Writing—review and editing; Maxim Ivanov, Conceptualization, Data curation, Formal analysis, Investigation, Visualization, Methodology, Writing—original draft, Writing—review and editing; Joseph Paul Parker, Formal analysis, Investigation, Visualization, Methodology; Stephen Buratowski, Data curation, Investigation; Sebastian Marquardt, Resources, Supervision, Funding acquisition, Investigation, Methodology, Writing—review and editing

### Author ORCIDs

Dorine Jeanne Mariëtte du Mee (iD) http://orcid.org/0000-0002-0037-7882
Maxim Ivanov (iD) http://orcid.org/0000-0001-7548-3316
Stephen Buratowski (iD) http://orcid.org/0000-0003-0440-3926
Sebastian Marquardt (iD) http://orcid.org/0000-0003-1709-2717

### Decision letter and Author response

Decision letter https://doi.org/10.7554/eLife.31989.019
Author response https://doi.org/10.7554/eLife.31989.020

## Additional files

### Supplementary files

• Supplementary file 1. Table S1 Yeast strains. Names, genotypes, purposes and origins of yeast strains used in the manuscript are indicated.
DOI: https://doi.org/10.7554/eLife.31989.015

• Supplementary file 2. Table S2 Oligonucleotides. Names, DNA sequences and purposes of oligonucleotides used in the manuscript are indicated.
DOI: https://doi.org/10.7554/eLife.31989.016

• Supplementary file 3. Table S3 Plasmids. Names, identifiers and purposes of plasmids used in the manuscript are indicated.
DOI: https://doi.org/10.7554/eLife.31989.017

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
