## [Decision Letter]

Thank you for submitting your article "Efficient Termination of Nuclear lncRNA Transcription Promotes Mitochondrial Genome Maintenance" for consideration by *eLife*. Your article has been reviewed by three peer reviewers, one of whom, Nick J Proudfoot (Reviewer #1), is a member of our Board of Reviewing Editors and the evaluation has been overseen by James Manley as the Senior Editor.

The reviewers have discussed the reviews with one another and the Reviewing Editor has drafted this decision to help you prepare a revised submission.

This paper describes, in *S. cerevisiae*, the case of a lncRNA (*CUT60*) that is positioned in tandem with an essential downstream mitochondrial protein coding gene *ATP16*. Remarkably, *CUT60* must be efficiently terminated to prevent transcriptional interference (TI) of *ATP16* with resultant loss of mitochondrial function. While TI is a well-established phenomenon for both yeast and mammalian genes, this study provides a dramatic example of this and extends mechanistic knowledge of the TI molecular mechanism. Furthermore it shows TI can also occur between a CUT type lncRNA and protein coding gene in yeast. Notably this has already been shown for SER3 in yeast as cited (Martens et al., 2004) and also in some mammalian lncRNA where pre-miRNA lncRNA fail to terminate without Drosha cleavage leading to down regulation of downstream protein coding genes by TI (Dhir et al., 2015). This later paper should also be cited.

Overall the data presented is thorough and convincing as far as it goes. Essentially the results show that insertion of URA3 into the *CUT60* locus disrupts termination. This causes read-through transcription into *ATP16* which blocks its expression and consequently causes a dramatic reduction in cell growth due to loss of mitochondrial function. Various further insertions of other CUTs as well as sequence dissection of the *CUT60* locus show that efficient termination is required to prevent *ATP16* downregulation by TI.

Two expert reviewers and the Reviewing Editor have considered your paper in detail. We feel that with further data to generalise the observed role of *CUT60* in protecting the tandem downstream protein coding gene *ATP16* from TI together with some tighter mechanistic analysis, then your paper will contain sufficient novel data to warrant *eLife* publication. Also the manuscript requires a substantial rewrite to make the data more accessible to non-specialists.

Specific points.

1) The ability of *CUT60* to terminate transcription reading into the unidirectional *ATP16* gene needs generalising. Other candidate examples of CUTs that act to block TI of downstream protein coding genes should be investigated genome wide. Presumably this class of CUT will have strong NNS termination signals. Analysis of existing NET-seq datasets could be used to identify these CUTs. Once these CUTs are identified one or more of them might be a good replacement candidate for *CUT60*.

Indeed it is unclear whether *CUT60* does not contain sequence that may be important for the ATP60 promoter since the actual replacements strategies failed to suppress totally the *CUT60* deletion.

2) The *CUT60* termination process/signals need fuller investigation. Does the lack of TI correlate with stronger termination signals? Nascent transcription analysis showing that TI caused by loss of *CUT60* termination is associated with read-through transcription into the *ATP16* promoter is required. E.g.by thioU-seq or NET-seq.

3) Trans mutants should be employed to decisively conclude that this is transcriptional interference. For instance nrd1 viable or attenuated mutants could be tested to establish whether they mimic the mitochondrial defects observed in *CUT60* mutant? Alternatively overexpression of NNS factors could enhance suppression (also by other cut sequences) to achieve similar levels of insulation? Similarly does the tethering the CrisprCas9dead on a cut60 deletion mutant help to suppress TI?

4) What is the nature or the URA3++ insert with enhanced termination? Why is its terminator now much stronger? Is this due to insertion of multiple NNS sites? Details of the exact nature of this terminator sequence is required. Also with the URA3 insertion into *CUT60*, the exact joins made in this homologous recombinant should be provided to understand why it is no longer efficiently terminating.

5) The manuscript needs to be rewritten to make it more accessible to a general reader. Maybe start by showing the mitochondrial defect when *CUT60* termination is compromised. The overly yeast specific genetic nomenclature should be simplified.

---

## [Author Response]

Specific points.1) The ability of CUT60 to terminate transcription reading into the unidirectional ATP16 gene needs generalising. Other candidate examples of CUTs that act to block TI of downstream protein coding genes should be investigated genome wide. Presumably this class of CUT will have strong NNS termination signals. Analysis of existing NET-seq datasets could be used to identify these CUTs. Once these CUTs are identified one or more of them might be a good replacement candidate for CUT60.

We addressed this comment by identifying 68 CUTs with characteristics similar to *CUT60,* now shown in the new Figure 3—source data 1. One of these is *CUT217,* which is better than the CUTs we had initially tested at promoting *ATP16* expression in place of *CUT60*.

Dr. Maxim Ivanov performed the computational analyses and is included as author of the revised manuscript.

Figure 3—source data 1 includes a column quantifying NNS-dependent termination by CUTs as “read-through index”. We calculated it by re-analysing Pol II PAR-CLIP data generated by Schaugency, Merran and Corden, 2014. This publication measures the effects of a conditional depletion allele of *NRD1* (Nrd1-Anchor Away). For clarity, we visualize this data for the subset of CUTs used for genomic replacements in the revised manuscript (Figure 3—figure supplement 2).

Figure 3—source data 1 includes the two columns, “Nrd1 occupancy” and “Nab3 occupancy” to assess binding of the NNS components Nrd1 and Nab3. These numbers are based on re-analyses of Nrd1 and Nab3 PAR-CLIP data described in Schulz et al., 2013.

Promoter directionality metrics in Figure 3—source data 1 are derived from NET-seq data (Marquardt et al., 2014).

We tested *CUT60* replacements with two new insertion strains informed by Figure 3—source data 1. As an additional control, we generated a replacement strain using the artificial CUT clone #78 that was identified by an in vivo SELEX approach for efficient NNS termination in Porrua et al., 2012.

We add a clarifying schematic describing our “scarless” approach for generating the replacement mutants in more detail Figure 3—figure supplement 1.

*ATP16* expression analysis of the new replacement strains is shown in the revised Figure 3. *ATP16* expression increases in cut60Δ::#78, and even more so in *cut60Δ::CUT217*. The analysis of additional CUT replacement strains generalize our findings by supporting the conclusions previously based on *CUT170*. CUT termination upstream of *ATP16* promotes *ATP16* expression. In combination with the new data supporting NNS termination of *CUT60* (comments 2-3) these data strengthen the role for efficient NNS termination upstream of *ATP16* to promote expression.

Indeed it is unclear whether CUT60 does not contain sequence that may be important for the ATP60 promoter since the actual replacements strategies failed to suppress totally the CUT60 deletion.

We believe this comment may result from our confusing layout of old Figure 1. We address this comment with our revised Figure 2. While it is correct that replacing *CUT60* with other CUTs does not fully restore *ATP16* expression, revised Figure 2 shows that *ATP16* expression does not rely on any sequences in *CUT60. ATP16* expression is equal comparing construct V1 (excluding *CUT60*) and constructs V2-4 (including *CUT60*). The most parsimonious explanation of our combined results is that the endogenous *ATP16* locus relays efficient upstream transcriptional termination to downstream initiation.

2) The CUT60 termination process/signals need fuller investigation. Does the lack of TI correlate with stronger termination signals? Nascent transcription analysis showing that TI caused by loss of CUT60 termination is associated with read-through transcription into the ATP16 promoter is required. E.g.by thioU-seq or NET-seq.

To address this comment we analysed nascent transcription, RNA bound to Pol II, as assayed by the PAR-CLIP method. We use published data comparing wild type vs mutations in Nrd1 described in Schaughency, Merran and Corden, 2014. We find that reduced activity of NNS results in read-through transcription across the *ATP16* promoter. In combination with our new results of reduced *ATP16* expression in Nrd1 mutants (see comment 3), these data support the conclusion that the lack of TI correlates with stronger *CUT60* termination, as the reviewers suggest.

These results appear as a read-through index >1 for *CUT60* in Figure 3—source data 1. A genome browser screenshot of differential Pol II PAR-CLIP signal also supports this conclusion, as Pol II PAR-CLIP signal is increased downstream of CUT60 after Nrd1 depletion (Figure 3—figure supplement 2).

We have expanded our analysis of termination signals. We present data measuring binding of Nrd1 and Nab3 by PAR-CLIP (Schulz et al., 2013) in Figure 3—source data 1, and for the CUTs used for *CUT60* replacement we add a plot of these data as Figure 3—figure supplement 3. We expanded our revised discussion, as these metrics do not fully resolve the question of what makes *CUT60* such an efficient terminator. The reviewers are correct to assume that *CUT60* scores highly in metrics associated with efficient NNS termination.

3) Trans mutants should be employed to decisively conclude that this is transcriptional interference. For instance nrd1 viable or attenuated mutants could be tested to establish whether they mimic the mitochondrial defects observed in CUT60 mutant? Alternatively overexpression of NNS factors could enhance suppression (also by other cut sequences) to achieve similar levels of insulation? Similarly does the tethering the CrisprCas9dead on a cut60 deletion mutant help to suppress TI?

Mutations in NNS factors are essential and so we addressed this point using two hypomorphic mutant alleles of Nrd1 *nrd1∆151-214* (Vasiljeva et al., 2008) and *nrd1-1* (Steinmetz and Brow, 1998).

Mutations in Nrd1 mimic yeast growth defects on non-fermentable carbon sources that we observe for *cut60Δ::URA3*. In addition, reduced Nrd1 function correlates with reduced *ATP16* expression, consistent with growth defects caused by inefficient *CUT60* termination.

These data support our mechanistic conclusions as well as the biological relevance of *CUT60* termination by an analysis of trans-acting termination factors. We include the data in our revised Figure 3.

We tethered CRISPR-dCas9 to *SUT129* in circuit II and followed TI repression by measuring YFP fluorescence. We used a vector with inducible gRNA expression based on Smith et al., 2016 Genome Biology. We found elevated YFP expression when a gRNA targeting *SUT129* was induced compared to the no-gRNA control and the non-induced control. However, the fold-change of YFP increase was lower than the effect of suppressing TI by mutations in elongation factors (Figure 4). While these data could offer support to the idea that gRNA-dCas9 can inhibit TI, we believe that these experiments do not add enough to warrant inclusion in the revised manuscript.

4) What is the nature or the URA3++ insert with enhanced termination? Why is its terminator now much stronger? Is this due to insertion of multiple NNS sites? Details of the exact nature of this terminator sequence is required. Also with the URA3 insertion into CUT60, the exact joins made in this homologous recombinant should be provided to understand why it is no longer efficiently terminating.

A revised schematic of the insertions, including joint sequences and the nature of the termination sequences, has been added as revised Figure 1—figure supplement 1 and Figure 3—figure supplement 3.

To investigate the question of relative termination strength we generated additional strains with increased PAS-dependent termination signals downstream of the *URA3* gene, *cut60Δ::URA3+* and *cut60Δ::URA3-Trp1-term*. As expected, *ATP16* expression is elevated compared to *cut60Δ::URA3* to equally high levels in *cut60Δ::URA3+, cut60Δ::URA3++* and *cut60Δ::URA3-Trp1-term*. We note that the quantitative increase of *ATP16* expression in *cut60Δ::URA3++* is reduced compared to the original Figure 3 strain mixup is responsible for this discrepancy and it is corrected in the revised Figure 3. The data is now based on three independent strategies to terminate *URA3* transcription. All our strategies support positive effects on *ATP16* expression by mediating transcriptional termination upstream.

5) The manuscript needs to be rewritten to make it more accessible to a general reader. Maybe start by showing the mitochondrial defect when CUT60 termination is compromised. The overly yeast specific genetic nomenclature should be simplified.

We addressed this comment by changing Figure 1 and Figure 2 of the manuscript and making the suggested text changes. Our revised text improves the balance between accessibility for the generalists while maintaining sufficient scientific accuracy in detail for the aficionados.